# The inverse relationship between dietary anthocyanidins consumption and frailty: Findings from the National Health and Nutrition Examination Survey

Xiaofeng Zhang¤, Pengpeng Huang¤, Zhenhua Jin¤, Yanfei Wu [ORCID]¤*

Center for Rehabilitation Medicine, Rehabilitation & Sports Medicine Research Institute of Zhejiang Province, Department of Rehabilitation Medicine, Zhejiang Provincial People's Hospital (Affiliated People's Hospital), Hangzhou Medical College, Hangzhou, Zhejiang, China

¤ Current address: No.158 Shangtang Road, Hangzhou, Zhejiang Province, China
* wuyanfei@hmc.edu.cn

## Abstract

### Objective

This study investigates the association between dietary anthocyanidins consumption and the risk of frailty in the general adult population using data from the National Health and Nutrition Examination Survey (NHANES).

### Methods

A cross-sectional analysis of NHANES data was conducted, examining the relationship between dietary anthocyanidins intake and frailty risk. Dose-response relationships and subgroup analyses were performed to provide a comprehensive understanding of the associations.

### Results

The results indicate a significant inverse relationship between the consumption of specific anthocyanidin types, including malvidin and delphinidin, as well as total anthocyanidins, and the odds of frailty. Particularly, individuals aged 41–60 years and males exhibited a notable protective effect of anthocyanidins against frailty. Dose-response analyses revealed a U-shaped relationship between total anthocyanidins intake and frailty risk, with an optimal consumption level of 33.322 mg per day.

### Conclusion

This pioneering study emphasizes the potential protective role of dietary anthocyanidins in mitigating frailty, especially among middle-aged individuals and males.

**Data availability statement:** All data files are available from the NHANES (National Health and Nutrition Examination Survey).

**Funding:** This study was financially supported by the Natural Science Foundation of Zhejiang Province in the form of a grant (LGF21H170005) received by XZ. This study was also financially supported by the Medical Science and Technology Project of Zhejiang Province in the form of grants (2022KY576 and 2023KY453) received by YW. This study was also financially supported by Zhejiang Traditional Chinese Medicine Administration in the form of a grant (2024ZL269) received by XZ. The funders had no role in study design, data collection and analysis, decision to publish, or preparation of the manuscript.

**Competing interests:** All authors disclosed no relevant relationships.

## Introduction

Frailty, identified as a geriatric syndrome, diminishes physiological reserve, escalating susceptibility to adverse health outcomes such as falls, disability, hospitalization, and mortality [1]. Its prevalence is approximately 10% among adults aged 50–64 and rises to 43.7% in those aged 65 or older, with an increasing occurrence with advancing age [2]. This condition places a substantial burden on individuals, families, and healthcare systems, projecting itself as a significant public health challenge in the context of population aging [3].

The pathophysiology of frailty is a multifaceted and intricate process involving diverse factors, including physical inactivity, chronic inflammation, micronutrient deficits, oxidative stress, hormonal dysregulation, and sarcopenia [4]. These factors interact synergistically, contributing to the decline of multiple physiological systems and disrupting homeostasis. Therefore, it is imperative to pinpoint modifiable factors capable of preventing or delaying the onset and progression of frailty, as this could markedly enhance the quality of life and overall well-being of individuals. Dietary interventions that possess the potential to modulate these underlying processes may offer benefits in terms of frailty prevention and treatment [5].

One of the potential dietary interventions is flavonoid, a natural polyphenolic compound widely distributed in plants. Flavonoid may mitigate frailty by modulating some of the underlying mechanisms, such as enhancing muscle function, reducing inflammation, and improving micronutrient status. A recent meta-analysis of randomized controlled trials revealed that flavonoid supplementation had beneficial effects on skeletal muscle mass and gait speed in middle-aged and older adults without exercise intervention, but there was no significant improvement in muscle strength [6]. Another cohort study suggested that higher intake of flavonoid was associated with lower risk of frailty over a 12-year follow-up period [7]. Notwithstanding, these studies are limited by factors such as a small sample size, reliance on a single dietary assessment, and a lack of detailed information on the individual components of flavonoids.

Anthocyanidins are a specific type of flavonoid that are responsible for the red, blue, and purple pigments found in fruits and vegetables like grapes, blueberries, blackberries, cherries, and eggplants. The most common forms of anthocyanidins are cyanidin, delphinidin, malvidin, pelargonidin, peonidin, and petunidin [8]. Anthocyanidins have various health benefits, such as antioxidant, anti-inflammatory, anticancer, anti-cardiovascular, and anti-neurodegenerative properties [9]. However, the current evidence on the relationship between anthocyanidins consumption and frailty remains unclear.

Therefore, the aim of this study was to examine the association between six major anthocyanidins and total anthocyanidins consumption and frailty in a large and representative sample of US adults aged 18 years and older, using data from the National Health and Nutrition Examination Survey (NHANES) 2007-2010 and 2017-2018. Previous research explored the relationship between dietary flavonoid intake and frailty in U.S. adults [10,11]. However, these investigations excluded younger adults under the age of 20 and did not

specifically assess the effects of individual anthocyanidin subtypes. Building upon this foundation, our study aimed to further explore the relationship between dietary anthocyanidin intake—including specific subtypes—and frailty in a nationally representative U.S. population. This study may provide new insights into the role of anthocyanidins in frailty prevention and management and have important implications for dietary recommendations and interventions for adults in different age groups.

## Materials and methods

### Data source and study population

Our data were sourced from the National Health and Nutrition Examination Survey (NHANES), a comprehensive and nationally representative health survey conducted by the Centers for Disease Control and Prevention (CDC) in the United States. Administered by the National Center for Health Statistics (NCHS), NHANES collects a diverse set of health-related data through detailed interviews and standardized physical examinations. All procedures associated with NHANES have received approval from the NCHS Research Ethics Review Board, adhering to established ethical standards. We utilized de-identified secondary data from the NHANES cycles 2007–2010 and 2017–2018, as dietary flavonoid intake data were only available for these years. The combined dataset included 29,883 participants, of whom 17,888 were aged 18 years or older. Following the exclusion of participants with incomplete data on anthocyanidins intake and frailty index, our analyses focused on 14,520 participants (Fig 1).

### Anthocyanidins consumption measurement

The measurement of anthocyanidins consumption was conducted using data from the USDA's Food and Nutrient Database for Dietary Studies (FNDDS) and NHANES for the periods 2007–2010 and 2017–2018. FNDDS provided estimates of flavonoid consumption for the US population, and these estimates were integrated with NHANES data obtained from laboratory tests, examinations, and questionnaires to explore the association between flavonoid intake and human health. The FNDDS's flavonoid database offers insights into the intake quantities of six major flavonoids and 29 subtypes, along with the overall quantity of flavonoids, for each participant on both the first and second days. To estimate usual intake more accurately and reduce day-to-day variability, the average anthocyanidin intake from the first and second 24-hour dietary recalls was calculated for each participant, as recommended for analyses using NHANES dietary data.

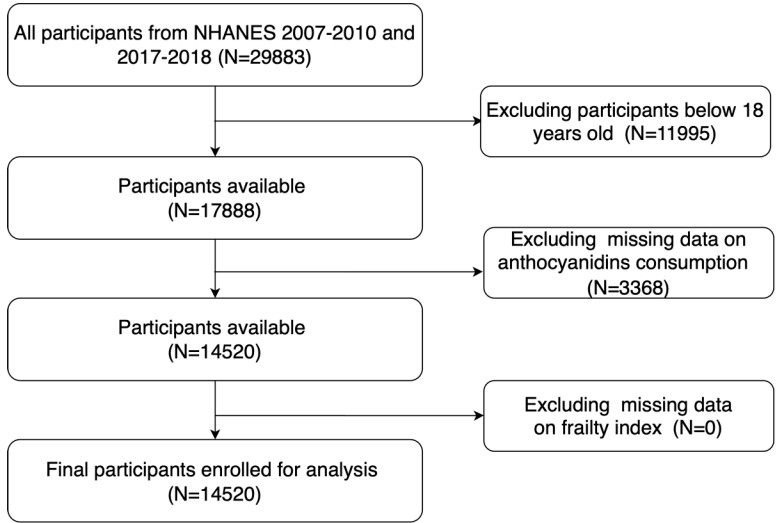

**Fig 1. Flowchart of participant selection.**

## Frailty index assessment

The frailty index serves as a comprehensive health measure, encompassing assessments of cognitive function, dependence, mental health, medical conditions, hospital utilization, overall health, physical performance, anthropometry, and laboratory results [12]. It consists of 49 factors evaluating participants' health status, all available in the NHANES database [13]. To derive the frailty index value, the participant's acquired deficits' number is divided by the total potential deficits [12]. This index ranges from 0 to 1, with higher values indicating increased levels of frailty. Frailty is specifically defined as a frailty index surpassing 0.21 [13]. Supplementary documentation provides detailed information on the specific variables included in the frailty index, highlighting its comprehensive nature and its ability to capture various aspects of health status (S1 Table).

## Covariates

This study assessed several covariates as potential confounders for each participant, including (1) sociodemographic information, such as sex (male, female), age (year), race (Mexican American, non-Hispanic black, non-Hispanic white, others), marital status (married/living with partner, never married, divorced/separated, widowed), education (under high school, high school, above high school), poverty income ratio levels(< 1.3, 1.3–3.49, ≥ 3.5) [14]; (2) body measurements, including body mass index (BMI, kg/m2); (3) life-style information, which included smoking status, drinking, physical activity, coffee consumption [15], and total daily energy intake (kcal/day); (4) laboratory tests, which included white blood cells count (WBC, 1000 cells/μL), hemoglobin (Hb, g/dL), platelets count (Plt, 1000 cells/μL, albumin (Alb, g/dL), creatinine (Cr, mg/dL), uric acid (UA, mg/dL), alanine transaminase (Alt, μ/L) and aspartate transaminase (Ast, μ/L) [16]; (5)common co-morbid conditions, including hypertension, hyperlipidemia, diabetes mellitus (DM), stroke, coronary heart disease, depressive symptom [17,18]. BMI was categories were as following 4 groups: underweight (BMI < 18.5), normal weight (18.5 ≤ BMI < 25), overweight (25 ≤ BMI < 30), and obesity (BMI ≥ 30) [19]. Smoking statuses are divided into three groups: never smokers (individuals who had smoked less than 100 cigarettes in their lifespan), former smokers (individuals who had smoked more than 100 cigarettes in their lifespan but were not currently smoking), and current smokers (those who were currently smoking). Alcohol drinking is divided into five categories: never drinkers (those who had drunk fewer than 12 times in their lifetime), former drinkers (those who had drunk more than 12 times in a year but were not currently drinking), mild drinkers (those who drank 1−2 drinks per day for women and 1−3 drinks per day for men), moderate drinkers (those who drank 2−3 drinks per day for women and 3−4 drinks per day for men), and heavy drinkers (those who drank more than 3 drinks per day for women and 4 drinks per day for men) [20]. Based on their weekly metabolic equivalent (MET) minutes, physical activity was divided into four groups: inactive (0 MET-min/week), low (1−499 MET-min/week), moderate (500−1000 MET-min/week), and high (>1000 MET-min/week). Hypertension is diagnosed when (1) an individual has been told by a physician or other health care provider that they have the condition, (2) has a history of using antihypertensive medication, (3) three separate measurements of systolic and diastolic blood pressure equal to or exceeding 140 mmHg and 90 mmHg, respectively [21]. The diagnostic criteria for hyperlipidemia include consist of a triglyceride concentration of 150 mg/dL or higher, a serum total cholesterol concentration of 200 mg/dL or higher, a low-density lipoprotein level of 130 mg/dL or greater, and a high-density lipoprotein level that is below 40 mg/dL in males or below 50 mg/dL in females, or the utilization of lipid-lowering medications [22]. The diagnostic criteria for diabetes include being told by a medical professional that one has the condition, having a glycosylated hemoglobin level of 6.5 mmol/L or above, a fasting blood glucose level of 7.0 mmol/L or above, or a history of taking anti-diabetic drugs [23]. The diagnostic criteria for coronary heart disease and stroke are based on whether the patient has ever been diagnosed with either condition. The nine questions on the Patient Health Questionnaire (PHQ-9) depression scale, which are based on symptoms of depression, were used to measure depressive symptoms. A total score between 0 and 27 is obtained by adding the scores on the nine items, which range from 0 (not at all) to 3 (almost every day) [24]. Clinically significant depression (CSD) was defined in this study as having a total PHQ-9 score of ≥ 10 [25].

## Statistical analyses

The participants were categorized into frail and non-frail groups based on their frailty index. Their intake of six anthocyanidin subtypes and total anthocyanidins was assessed, with further categorization into three groups for each anthocyanidin type. Group 1 comprised individuals with no detectable intake, while Group 2 and Group 3 included those with intake below and above or equal to the median, respectively. Descriptive statistics were used to summarize participant characteristics by frailty status. Continuous variables were presented as means with standard deviations (mean ± SD), while categorical variables were reported as unweighted counts with survey-weighted percentages.

To compare characteristics between frailty groups, statistical tests were selected based on variable type and distribution. Levene's test was first conducted to assess the homogeneity of variances for all continuous variables. If the assumption of equal variances was met ($p \geq 0.05$), a survey-weighted Student's t-test was applied. If the assumption was violated ($p < 0.05$), a survey-weighted Welch's t-test was used instead. For categorical variables, survey-weighted Rao–Scott corrected chi-square tests were conducted. In addition to p-values, effect sizes were calculated to evaluate the magnitude of group differences: Cohen's d was reported for continuous variables using unpooled standard deviations, while Cramer's V was used for categorical variables. Effect sizes were interpreted using conventional thresholds: for Cohen's d, 0.2 was considered small, 0.5 medium, and 0.8 large; for Cramer's V, 0.1 was considered small, 0.3 medium, and 0.5 large [26]. For categorical variables with statistically significant associations ($p < 0.05$) and more than two levels, standardized residual analyses were performed using unweighted contingency tables to identify specific category combinations contributing most to the overall chi-square statistic.

Four models of binary logistic regression analyses were conducted to explore the relationship between the consumption of the six main anthocyanidin categories, total anthocyanidins, and the incidence of frailty. The crude model, the first in the series, did not adjust for any confounding factors. Model 1 incorporated adjustments for age, sex, and race, while Model 2 further adjusted for education, marital status, poverty-income ratio, BMI, drinking status, smoking, caffeine consumption, total dietary energy intake, and physical activity, building upon Model 1. Model 3 extended adjustments to include white blood cells, hemoglobin, platelets, albumin, creatinine, uric acid, alanine transaminase, aspartate transaminase, hypertension, hyperlipidemia, diabetes, coronary heart disease, depressive symptoms, and stroke, in addition to the factors considered in Model 2. In Model 3, a restricted cubic spline curve with three knots at the 10th, 50th, and 90th percentiles of anthocyanidin intake was employed to explore the dose-response relationship between anthocyanidins consumption and odds ratios (ORs) along with 95% confidence intervals (CIs) of frailty.

Due to the complex survey design of the NHANES data, traditional model selection criteria such as the Akaike Information Criterion (AIC) or the Bayesian Information Criterion (BIC) were not applicable [27]. Instead, to assess improvements in model performance with the sequential inclusion of covariates, McFadden's Pseudo R² values were calculated for each anthocyanidin subtype. The crude model (Model 0), which included only the anthocyanidin intake variable, was treated as the null (reference) model. Models 1 through 3 were treated as progressively adjusted full models. McFadden's Pseudo R² was computed as:

Pseudo R² = 1 - (Residual Deviance of Full Model/ Residual Deviance of Null Model) [28].

Pseudo R² values < 0.1 suggest poor fit, values between 0.1–0.3 suggest modest fit, values from 0.3–0.5 indicate moderate fit, and values > 0.5 reflect strong model fit.

To generate nationally representative estimates, dietary weight was applied (wtdr2d). Furthermore, a subgroup analysis by age and sex was conducted to examine the consistency of the association between anthocyanidins intake and frailty risk across different age groups and sexes in the adult population. To ensure the robustness of our results, we performed an additional assessment by excluding participants with stroke, depressive symptoms, and coronary heart disease. Statistical analyses were executed using R software (version 4.2.0). The significance level for all tests was set at 0.05, and p-values were deemed statistically significant if ≤ 0.05.

## Results

### Characteristics of the participants at baseline

Table 1 displays the baseline characteristics of participants, categorized as either frail or non-frail based on their frailty index. The study encompassed 14,520 eligible participants with an average age of 46.42 years, of whom 52.66% were female. Among these participants, 11,182 (82%) were non-frail, while 3,338 (18%) were classified as frail. The weighted population estimates were approximately 191,053,521 non-frail adults, 41,031,334 frail adults, and 232,084,855 individuals in total, reflecting the national representation of the U.S. adult population after applying NHANES sampling weights.

A comparison between non-frail and frail individuals revealed notable differences. Frail participants, in contrast to their non-frail counterparts, tended to be older, predominantly female, non-Hispanic black, divorced/separated/widowed, less educated, economically disadvantaged, less physically active, current or former smokers, non-drinkers, and either underweight or obese. Additionally, frail individuals exhibited patterns of consuming less dietary energy and more decaffeinated coffee. They were also more likely to exhibit depressive symptoms; elevated white blood cell and platelet counts, uric acid, creatinine, and aspartate aminotransferase levels; reduced hemoglobin and alanine aminotransferase levels; and a higher prevalence of comorbidities, including hypertension, hyperlipidemia, diabetes, stroke, and coronary heart disease.

To facilitate interpretation, effect sizes (Cohen's d or Cramer's V) were reported for all comparisons. Several variables demonstrated medium to large effect sizes, indicating substantial differences between the frail and non-frail groups. Notably, age (Cohen's d = –0.63), physical activity (Cohen's d = 0.55), education level (Cohen's d = 0.51), and race/ethnicity (Cramer's V = 0.39) exhibited the largest effect sizes (Table 1).

### Relationship between anthocyanidins consumption and frailty

Table 2 illustrates the association between specific anthocyanidins and total anthocyanidins intake with frailty. Increased consumption of malvidin, delphinidin, and total anthocyanidins was significantly and inversely linked to the odds ratio of frailty across all models. Participants with malvidin intake-above the median exhibited ORs of 0.64 (95% CI: 0.55–0.75) and 0.49 (95% CI: 0.42–0.57) for frailty in the crude and Model 1, respectively, and ORs of 0.69 (95% CI: 0.53–0.88) in Model 2 and 0.67 (95% CI: 0.49–0.92) in Model 3, respectively. Similarly, participants with above median delphinidin consumption showed ORs of 0.62 (95% CI: 0.54–0.73) in the crude model, 0.48 (95% CI: 0.41–0.57) in Model 1, 0.74 (95% CI: 0.57–0.96) in Model 2, and 0.73 (95% CI: 0.54–0.99) in Model 3, respectively. Additionally, the OR for above-median total anthocyanidins intake in the crude model was 0.6 (95% CI: 0.48–0.74), decreasing to 0.41 (95% CI: 0.32–0.51) in Model 1, and rising to 0.61 (95% CI: 0.44–0.85) and 0.59 (95% CI: 0.39–0.90) in Models 2 and 3, respectively.

The intake of individual anthocyanidins and total anthocyanidins was not significantly associated with frailty overall, except for delphinidin, which exhibited an OR of 0.97 (95% CI: 0.96, 0.99), 0.96 (95% CI: 0.94, 0.98), 0.96 (95% CI: 0.94, 0.99), and 0.97 (95% CI: 0.94, 1.00) in the crude model, Model 1, Model 2, and Model 3, respectively.

The Pseudo R² values increased progressively with each stepwise adjustment, indicating improved model fit. For example, in the analysis of total anthocyanidin intake, Model 1 yielded a Pseudo R² of 0.092, which increased substantially to 0.562 in Model 2. The fully adjusted Model 3 demonstrated the best model performance, with a Pseudo R² of 0.695. A similar pattern was observed for all individual anthocyanidin subtypes (e.g., cyanidin, delphinidin, malvidin), with the Pseudo R² values in the fully adjusted models consistently ranging from approximately 0.694 to 0.696.

To assess the robustness of our findings, additional analyses were conducted by excluding participants with stroke, coronary heart disease, and depressive symptoms. Interestingly, a significant inverse association with frailty was observed for higher (above-median) intake of malvidin, petunidin, delphinidin, and total anthocyanidins, whereas no such association was found for cyanidin, peonidin, or pelargonidin. Furthermore, in the fully adjusted models examining the overall association between individual anthocyanidins and frailty, significant relationships persisted for petunidin and delphinidin (S2 Table).

**Table 1. Weighted characteristics of participants by frailty status.**

| Characteristic | N[1] | No frailty (n = 11182)[1] | Frailty (n = 3338)[1] | Effect size[2] | P Value |
|---|---|---|---|---|---|
| Age (years) | 14,520 | 44.44 ± 16.84 | 55.68 ± 17.13 | Cohen's d = −0.63 | <0.001 |
| Sex | 14,520 | | | Cohen's d = 0.3 | <0.001 |
| Female | | 5,605 (50.76%) | 1,969 (61.47%) | | |
| Male | | 5,577 (49.24%) | 1,369 (38.53%) | | |
| Race | 14,520 | | | Cramer's V = 0.39 | <0.001 |
| Non-Hispanic White | | 4,990 (67.47%) | 1,531 (63.35%) | | |
| Non-Hispanic Black | | 2,187 (10.45%) | 844 (16.72%) | | |
| Mexican American | | 1,911 (9.02%) | 429 (7.27%) | | |
| Other Race | | 2,094 (13.05%) | 534 (12.67%) | | |
| Marital status | 14,520 | | | Cohen's d = −0.11 | <0.001 |
| Married/living with partner | | 6,708 (62.21%) | 1,642 (54.11%) | | |
| Never married | | 1,950 (19.75%) | 385 (12.03%) | | |
| Widowed | | 628 (3.89%) | 533 (13.43%) | | |
| Divorced/separated | | 1,371 (10.89%) | 635 (16.47%) | | |
| Unknow | | 525 (3.26%) | 143 (3.96%) | | |
| Education | 14,520 | | | Cohen's d = 0.51 | <0.001 |
| High school | | 4,220 (34.34%) | 1,537 (47.27%) | | |
| Below high school | | 956 (3.99%) | 500 (9.11%) | | |
| Above high school | | 5,992 (61.58%) | 1,294 (43.55%) | | |
| Unknow | | 14 (0.09%) | 7 (0.08%) | | |
| Poverty income ratio | 14,520 | | | Cohen's d = −0.05 | <0.001 |
| ≤ 1.3 | | 2,812 (17.17%) | 1,273 (32.29%) | | |
| > 1.3 and ≤3.5 | | 3,905 (31.98%) | 1,207 (33.66%) | | |
| > 3.5 | | 3,434 (43.18%) | 523 (25.56%) | | |
| Unknow | | 1,031 (7.68%) | 335 (8.50%) | | |
| Physical activity | 14,520 | | | Cohen's d = 0.55 | <0.001 |
| High | | 6,277 (60.72%) | 1,109 (36.19%) | | |
| Moderate | | 1,195 (10.64%) | 259 (8.52%) | | |
| Low | | 1,277 (11.15%) | 436 (13.44%) | | |
| Unknow | | 2,433 (17.49%) | 1,534 (41.85%) | | |
| Smoking status | 14,520 | | | Cohen's d = −0.27 | <0.001 |
| Never | | 6,341 (58.25%) | 1,413 (41.98%) | | |
| Former | | 2,484 (22.62%) | 1,033 (29.80%) | | |
| Current | | 1,992 (17%) | 795 (25.94%) | | |
| Unknow | | 365 (2.13%) | 97 (2.28%) | | |
| Drinking status | 14,520 | | | Cohen's d = −0.22 | <0.001 |
| Heavy drinker | | 1,502 (15.53%) | 255 (9.07%) | | |
| Moderate drinker | | 1,696 (16.28%) | 329 (10.78%) | | |
| No drinker | | 7,984 (68.19%) | 2,754 (80.16%) | | |
| BMI | 14,520 | | | Cohen's d = 0.09 | <0.001 |
| Underweight | | 180 (1.59%) | 60 (1.98%) | | |
| Normal | | 3,231 (30.68%) | 544 (15.25%) | | |
| Overweight | | 3,788 (32.67%) | 940 (26.86%) | | |
| Obese | | 3,931 (34.78%) | 1,679 (52.68%) | | |
| Unknow | | 52 (0.28%) | 115 (3.23%) | | |

*(Continued)*

**Table 1.** (Continued)

| Characteristic | N¹ | No frailty (n = 11182)¹ | Frailty (n = 3338)¹ | Effect size² | P Value |
|---|---|---|---|---|---|
| Caffeinated coffee (g/day) | 14,518 | 281.17±471.66 | 296.60±617.90 | Cramer's V = 0.07 | 0.3 |
| Decaffeinated coffee (g/day) | 14,518 | 31.01±160.15 | 45.87±258.59 | Cramer's V = 0.07 | 0.029 |
| Energy consumption (kcal/day) | 14,520 | 2,124.54±837.03 | 1,879.44±804.31 | Cramer's V = 0.19 | <0.001 |
| Depressive symptom | 14,520 | | | Cohen's d = −0.01 | <0.001 |
| No | | 10,255 (91.93%) | 2,140 (62.85%) | | |
| Yes | | 323 (2.98%) | 950 (30.79%) | | |
| Unknow | | 604 (5.09%) | 248 (6.36%) | | |
| White blood cells count (1000 cells/µL) | 13,925 | 7.19±2.08 | 7.93±6.56 | Cramer's V = 0.13 | <0.001 |
| Hemoglobin (g/dL) | 13,926 | 14.36±1.40 | 13.69±1.67 | Cramer's V = 0.16 | <0.001 |
| Platelets count (1000 cells/µL) | 13,926 | 249.26±61.79 | 257.71±81.33 | Cramer's V = 0.24 | 0.003 |
| Albumin (g/dL) | 13,752 | 4.25±0.33 | 4.04±0.37 | Cramer's V = 0.13 | <0.001 |
| Creatinine (mg/dL) | 13,750 | 0.86±0.27 | 0.96±0.59 | Cramer's V = 0.11 | <0.001 |
| Uric acid (mg/dL) | 13,747 | 5.38±1.39 | 5.69±1.58 | Cramer's V = 0.19 | <0.001 |
| Alanine transaminase (µ/L) | 13,746 | 24.91±17.54 | 23.69±17.58 | Cohen's d = −0.07 | 0.031 |
| Aspartate transaminase (µ/L) | 13,730 | 24.57±13.18 | 25.19±15.84 | Cohen's d = −0.05 | 0.11 |
| Hypertension | 14,520 | | | Cramer's V = 0.31 | <0.001 |
| Yes | | 3,784 (28.89%) | 2,336 (68.58%) | | |
| No | | 7,397 (71.10%) | 1,002 (31.42%) | | |
| Unknow | | 1 (0.01%) | 0 (0.00%) | | |
| Hyperlipidemia | 14,520 | | | Cramer's V = 0.12 | <0.001 |
| Yes | | 7,483 (65.35%) | 2,659 (80.46%) | | |
| No | | 3,699 (34.65%) | 677 (19.52%) | | |
| Unknow | | 0 (0.00%) | 2 (0.02%) | | |
| Diabetes | 14,520 | | | Cramer's V = 0.33 | <0.001 |
| Yes | | 860 (5.92%) | 1,151 (7.93%) | | |
| No | | 10,193 (93.47%) | 2,163 (67.73%) | | |
| Unknow | | 129 (1.17%) | 24 (1.08%) | | |
| Stroke | 14,520 | | | Cramer's V = 0.25 | <0.001 |
| Yes | | 148 (1.00%) | 441 (12.43%) | | |
| No | | 10,506 (95.72%) | 2,740 (83.04%) | | |
| Unknow | | 528 (3.28%) | 157 (4.53%) | | |
| Coronary heart disease | 14,520 | | | Cramer's V = 0.22 | <0.001 |
| Yes | | 198 (1.37%) | 418 (12.88%) | | |
| No | | 10,442 (95.26%) | 2,749 (82.64%) | | |
| Unknow | | 542 (3.37%) | 171 (4.48%) | | |

¹N represents the total unweighted sample size, and n represents the unweighted number of participants in each frailty group. Percentages are based on NHANES survey-weighted analyses. The weighted population estimates were approximately **191,053,521** (no frailty), **41,031,334** (frailty), and **232,084,855** (total).

²Levene's test was used to assess homogeneity of variances for continuous variables to determine whether Student's t-test or Welch's t-test was appropriate. The Rao-Scott F-adjusted chi-square test was used for categorical variables. Effect sizes are reported as Cohen's d for continuous variables and Cramer's V for categorical variables.

**Table 2. The relationship between anthocyanidins consumption and frailty.**

| Characteristic | Crude model OR1 | 95% CI1 | P-value | Model1[a] OR1 | 95% CI1 | p-value | Pseudo R² | Model2[b] OR1 | 95% CI1 | P-value | Pseudo R² | Model3[c] OR1 | 95% CI1 | P-value | Pseudo R² |
|---|---|---|---|---|---|---|---|---|---|---|---|---|---|---|---|
| Cyanidin[d] | | | | | | | 0.09 | | | | 0.563 | | | | 0.696 |
| group1 | — | — | | — | — | | | — | — | | | — | — | | |
| group2 | 0.76 | 0.63, 0.91 | 0.004 | 0.61 | 0.50, 0.74 | <0.001 | | 0.69 | 0.49, 0.97 | 0.036 | | 0.67 | 0.41, 1.08 | 0.089 | |
| group3 | 0.71 | 0.59, 0.85 | <0.001 | 0.48 | 0.39, 0.58 | <0.001 | | 0.71 | 0.52, 0.98 | 0.037 | | 0.73 | 0.48, 1.10 | 0.12 | |
| Overall[e] | 0.99 | 0.98,1.00 | 0.001 | 0.98 | 0.97, 0.99 | <0.001 | | 1.00 | 0.99, 1.01 | 0.8 | | 1.00 | 1.0, 1.01 | 0.4 | |
| Malvidin[d] | | | | | | | 0.09 | | | | 0.562 | | | | 0.695 |
| group1 | — | — | | — | — | | | — | — | | | — | — | | |
| group2 | 0.71 | 0.59, 0.84 | <0.001 | 0.62 | 0.52, 0.74 | <0.001 | | 0.89 | 0.66, 1.20 | 0.4 | | 0.96 | 0.69, 1.33 | 0.8 | |
| group3 | 0.64 | 0.55, 0.75 | <0.001 | 0.49 | 0.42, 0.57 | <0.001 | | 0.69 | 0.53, 0.88 | 0.006 | | 0.67 | 0.49, 0.92 | 0.017 | |
| Overall[e] | 0.99 | 0.98, 1.0 | 0.001 | 0.98 | 0.97, 0.99 | <0.001 | | 0.99 | 0.98, 1.00 | 0.037 | | 0.99 | 0.97, 1.00 | 0.10 | |
| Peonidin[d] | | | | | | | 0.088 | | | | 0.562 | | | | 0.694 |
| group1 | — | — | | — | — | | | — | — | | | — | — | | |
| group2 | 0.87 | 0.76, 1.01 | 0.062 | 0.74 | 0.63, 0.86 | <0.001 | | 0.94 | 0.69, 1.27 | 0.6 | | 0.92 | 0.61, 1.40 | 0.7 | |
| group3 | 0.70 | 0.60, 0.81 | <0.001 | 0.54 | 0.46, 0.63 | <0.001 | | 0.77 | 0.61, 0.97 | 0.029 | | 0.79 | 0.58, 1.07 | 0.11 | |
| Overall[e] | 1.00 | 1.0, 1.01 | 0.7 | 1.00 | 0.99, 1.01 | >0.9 | | 1.01 | 1.00, 1.02 | 0.2 | | 1.00 | 1.0, 1.01 | 0.5 | |
| Petunidin[d] | | | | | | | 0.088 | | | | 0.562 | | | | 0.694 |
| group1 | — | — | | — | — | | | — | — | | | — | — | | |
| group2 | 0.77 | 0.65, 0.91 | 0.003 | 0.70 | 0.58, 0.84 | <0.001 | | 0.89 | 0.64, 1.23 | 0.4 | | 0.95 | 0.61, 1.48 | 0.8 | |
| group3 | 0.62 | 0.53, 0.72 | <0.001 | 0.50 | 0.43, 0.59 | <0.001 | | 0.72 | 0.55, 0.93 | 0.016 | | 0.77 | 0.56, 1.05 | 0.093 | |
| Overall[e] | 0.97 | 0.94, 0.99 | 0.010 | 0.95 | 0.92, 0.98 | <0.001 | | 0.95 | 0.92, 0.99 | 0.008 | | 0.97 | 0.93, 1.01 | 0.10 | |
| Pelargonidin[d] | | | | | | | 0.085 | | | | 0.563 | | | | 0.695 |
| group1 | — | — | | — | — | | | — | — | | | — | — | | |
| group2 | 0.98 | 0.83, 1.17 | 0.9 | 0.84 | 0.70, 1.00 | 0.056 | | 1.05 | 0.76, 1.44 | 0.8 | | 0.96 | 0.65, 1.42 | 0.8 | |
| group3 | 0.76 | 0.65, 0.88 | <0.001 | 0.62 | 0.53, 0.73 | <0.001 | | 0.89 | 0.71, 1.10 | 0.3 | | 0.94 | 0.70, 1.27 | 0.7 | |
| Overall[e] | 0.98 | 0.97, 1.00 | 0.022 | 0.97 | 0.96, 0.99 | 0.009 | | 0.99 | 0.96, 1.02 | 0.4 | | 1.00 | 0.97, 1.04 | 0.8 | |
| Delphinidin[d] | | | | | | | 0.09 | | | | 0.562 | | | | 0.694 |
| group1 | — | — | | — | — | | | — | — | | | — | — | | |
| group2 | 0.78 | 0.66, 0.92 | 0.003 | 0.63 | 0.53, 0.75 | <0.001 | | 0.84 | 0.64, 1.12 | 0.2 | | 0.81 | 0.57, 1.16 | 0.2 | |
| group3 | 0.62 | 0.54, 0.73 | <0.001 | 0.48 | 0.41, 0.57 | <0.001 | | 0.74 | 0.57, 0.96 | 0.025 | | 0.73 | 0.54, 0.99 | 0.045 | |

*(Continued)*

**Table 2.** (Continued)

| Characteristic | Crude model | | | Model1[a] | | | | Model2[b] | | | | Model3[c] | | | |
|---|---|---|---|---|---|---|---|---|---|---|---|---|---|---|---|
| | OR1 | 95% CI1 | P-value | OR1 | 95% CI1 | p-value | Pseudo R² | OR1 | 95% CI1 | P-value | Pseudo R² | OR1 | 95% CI1 | P-value | Pseudo R² |
| Overall[e] | 0.97 | 0.96, 0.99 | 0.002 | 0.96 | 0.94, 0.98 | <0.001 | | 0.96 | 0.94, 0.99 | 0.004 | | 0.97 | 0.94, 1.00 | 0.035 | |
| Total anthocyanidins[e] | | | | | | | 0.092 | | | | 0.562 | | | | 0.695 |
| group1 | — | — | | — | — | | | — | — | | | — | — | | |
| group2 | 0.86 | 0.72, 1.02 | 0.075 | 0.68 | 0.57, 0.81 | <0.001 | | 0.81 | 0.60, 1.10 | 0.2 | | 0.83 | 0.53, 1.29 | 0.4 | |
| group3 | 0.60 | 0.48, 0.74 | <0.001 | 0.41 | 0.32, 0.51 | <0.001 | | 0.61 | 0.44, 0.85 | 0.007 | | 0.59 | 0.39, 0.90 | 0.020 | |
| Overall[e] | 1.00 | 0.99, 1.00 | 0.055 | 0.99 | 0.99, 1.00 | 0.010 | | 1.00 | 0.99, 1.00 | 0.6 | | 1.00 | 0.99, 1.00 | 0.7 | |

**Abbreviations:** OR, odds ratio; CI, confidence interval.

[a]Model 1 was adjusted for age, sex (male and female) and race (Mexican American, non-Hispanic black, non-Hispanic white, other races).

[b]Model 2 was additionally adjusted for education (below high school, high school, above high school), marital status (married/living with partner, never married, divorced/separated, widowed), poverty-income ratio (continuous), body mass index(BMI) (<18.5, 18.5≤BMI<25, 25≤BMI<30, ≥30, kg/m2), drinking status (no drinker, moderate drinker, heavy drinker), smoking (never, former, current), caffeine consumption(continuous, g/day), total dietary energy intake(continuous, kcal/day), and physical activity(low, moderate, high);

[c]Model 3 was additionally adjusted for white blood cells count (1000 cells/μL), hemoglobin (g/dL), platelets count (1000 cells/μL), albumin (g/dL), creatinine (mg/dL), uric acid (mg/dL), alanine transaminase (μ/L) and aspartate transaminase (μ/L), hypertension, hyperlipidemia, depressive symptom, diabetes, coronary heart disease, and stroke.

[d]The consumption of each type of anthocyanidin was categorized into three groups: Group 1 (no consumption, i.e., intake=0), Group 2 (intake below the median), and Group 3 (intake at or above the median).

[e]Analysis of each anthocyanidin subtype and total anthocyanidins as continuous intake variables.

## Subgroup analysis of the relationship between anthocyanidins consumption and frailty

Table 3 illustrates the association between anthocyanidin consumption and frailty risk across different age groups and sexes. Analyzing data by age groups revealed that higher cyanidin intake was significantly linked to reduced frailty risk in both group 2 (OR = 0.526; 95% CI: 0.375–0.738) and group 3 (OR = 0.533; 95% CI: 0.376–0.757) among participants aged 41–60, exhibiting a noteworthy trend (P = 0.002). In the 41–60 age group, an elevated intake of malvidin at or above the median correlated with a decreased frailty risk (OR = 0.514; 95% CI: 0.359–0.735). Additionally, both below-median (OR = 0.721; 95% CI: 0.531–0.980) and at or above-median (OR = 0.538; 95% CI: 0.389–0.744) peonidin consumption were significantly associated with a lower odds ratio of frailty in adults aged 41–60. Similarly, higher consumption of petunidin (OR = 0.502; 95% CI: 0.355–0.712) and pelargonidin (OR = 0.708; 95% CI: 0.513–0.977) was linked to a reduced risk of frailty in individuals aged 41–60. Delphinidin intake showed a significant trend toward decreasing frailty risk in both group 2 (OR = 0.707; 95% CI: 0.521–0.959) and group 3 (OR = 0.489; 95% CI: 0.350–0.683) among individuals aged 41–60. Similarly, the odds ratio of frailty was 0.567 (95% CI: 0.404–0.796) for participants consuming less than the median of total anthocyanidins, and decreased further to 0.452 (95% CI: 0.318–0.644) among adults aged 41–60 years whose intake was at or above the median. However, the relationship between other types of anthocyanidin consumption and frailty in different age groups was insignificant.

When examining data by sex, it was observed that the consumption of all anthocyanidin types, except for peonidin and pelargonidin, was associated with a lower risk of frailty in males. Specifically, higher intake of malvidin, petunidin, and delphinidin significantly correlated with lower odds of frailty in males, with odds ratios (OR) of 0.724 (95% CI: 0.525-0.998),

**Table 3. Association between anthocyanidin consumption and frailty risk in different subgroups.**

| | Subgroups | Group 1[a] | Group 2[a] | Group 3[a] | P for trend | P for interaction |
|---|---|---|---|---|---|---|
| Cyanidin | 18-40 Years Old | reference | 0.79(0.508, 1.229) | 1.119(0.689, 1.817) | 0.62 | < 0.001 |
| | 41-60 Years Old | reference | 0.526(0.375, 0.738) | 0.533(0.376, 0.757) | 0.002 | |
| | ≥ 60 Years Old | reference | 0.996(0.677, 1.466) | 1.441(0.982, 2.114) | 0.013 | |
| | Female | reference | 0.849(0.630, 1.144) | 0.891(0.654, 1.215) | 0.602 | 0.022 |
| | Male | reference | 0.594(0.437, 0.806) | 0.941(0.692, 1.279) | 0.78 | |
| Malvidin | 18-40 Years Old | reference | 0.96(0.605, 1.524) | 1.172(0.707, 1.945) | 0.63 | < 0.001 |
| | 41-60 Years Old | reference | 0.769(0.563, 1.051) | 0.514(0.359, 0.735) | <0.001 | |
| | ≥ 60 Years Old | reference | 1.027(0.746, 1.413) | 1.009(0.740, 1.377) | 0.933 | |
| | Female | reference | 0.991(0.762, 1.289) | 0.86(0.651, 1.135) | 0.322 | 0.182 |
| | Male | reference | 0.92(0.684, 1.237) | 0.724(0.525, 0.998) | 0.055 | |
| Peonidin | 18-40 Years Old | reference | 0.845(0.553, 1.289) | 0.917(0.584, 1.441) | 0.637 | < 0.001 |
| | 41-60 Years Old | reference | 0.721(0.531, 0.980) | 0.538(0.389, 0.744) | <0.001 | |
| | ≥ 60 Years Old | reference | 1.081(0.789, 1.481) | 1.161(0.853, 1.582) | 0.342 | |
| | Female | reference | 0.863(0.661, 1.126) | 0.894(0.684, 1.170) | 0.427 | 0.052 |
| | Male | reference | 1.00(0.763, 1.310) | 0.759(0.568, 1.014) | 0.074 | |
| Petunidin | 18-40 Years Old | reference | 0.868(0.553, 1.363) | 0.98(0.599, 1.604) | 0.798 | < 0.001 |
| | 41-60 Years Old | reference | 0.794(0.583, 1.080) | 0.502(0.355, 0.712) | <0.001 | |
| | ≥ 60 Years Old | reference | 0.998(0.732, 1.360) | 1.089(0.800, 1.482) | 0.619 | |
| | Female | reference | 0.919(0.708, 1.194) | 0.901(0.688, 1.179) | 0.412 | 0.079 |
| | Male | reference | 0.888(0.670, 1.177) | 0.673(0.491, 0.923) | 0.016 | |
| Pelargonidin | 18-40 Years Old | reference | 0.74(0.477, 1.148) | 0.761(0.482, 1.202) | 0.173 | 0.097 |
| | 41-60 Years Old | reference | 0.821(0.606, 1.112) | 0.708(0.513, 0.977) | 0.031 | |
| | ≥ 60 Years Old | reference | 1.009(0.750, 1.356) | 0.952(0.705, 1.287) | 0.764 | |
| | Female | reference | 0.877(0.679, 1.132) | 0.838(0.645, 1.088) | 0.17 | 0.575 |
| | Male | reference | 0.913(0.698, 1.195) | 0.774(0.579, 1.033) | 0.086 | |
| Delphinidin | 18-40 Years Old | reference | 0.91(0.588, 1.408) | 1.042(0.651, 1.668) | 0.962 | < 0.001 |
| | 41-60 Years Old | reference | 0.707(0.521, 0.959) | 0.489(0.350, 0.683) | <0.001 | |
| | ≥ 60 Years Old | reference | 0.991(0.733, 1.340) | 1.093(0.801, 1.492) | 0.599 | |
| | Female | reference | 0.899(0.696, 1.162) | 0.885(0.677, 1.157) | 0.345 | 0.121 |
| | Male | reference | 0.894(0.679, 1.178) | 0.702(0.519, 0.950) | 0.024 | |
| Total anthocyanidins | 18-40 Years Old | reference | 1.055(0.676, 1.645) | 0.932(0.567, 1.531) | 0.761 | < 0.0001 |
| | 41-60 Years Old | reference | 0.567(0.404, 0.796) | 0.452(0.318, 0.644) | <0.001 | |
| | ≥ 60 Years Old | reference | 1.149(0.775, 1.703) | 1.288(0.867, 1.913) | 0.188 | |
| | Female | reference | 0.931(0.686, 1.264) | 0.827(0.605, 1.131) | 0.197 | 0.407 |
| | Male | reference | 0.71(0.526, 0.959) | 0.747(0.544, 1.025) | 0.119 | |

[a]The consumption of each anthocyanidins was divided into three categories. Group 1 included individuals who did not consume any quantity of anthocyanidins, represented by a consumption value of 0. Group 2 was composed of individuals whose intake was below the median, while Group 3 encompassed individuals whose intake was at or above the median.

P for trend was derived from survey-weighted logistic regression models treating intake category as an ordinal variable. P for interaction was obtained from models including a cross-product term between anthocyanidin intake and the subgroup variable, accounting for NHANES complex survey design.

0.673 (95% CI: 0.491-0.923), and 0.702 (95% CI: 0.519-0.950), respectively. Moreover, consumption of cyanidin (OR, 0.594; 95% CI: 0.437-0.806) and total anthocyanidins (OR, 0.71; 95% CI: 0.526-0.959) below the median was associated with a lower risk of frailty in males.

## Dose-response relationships between anthocyanidins intake and frailty risk

In the investigation of dose-response relationships between anthocyanidin intake and frailty risk, restricted cubic spline analyses revealed a nonlinear inverse (U-shaped) association with total anthocyanidins intake and frailty (P < 0.001). The nadir of frailty odds ratio was 0.587 (95% CI, 0.452–0.763) at an intake of 33.322 mg per day for total anthocyanidins (Fig 2a). Beyond this threshold, the trend gradually reversed. However, the intake of malvidin (Fig 2b, P = 0.070) and delphinidin (Fig 2c, P = 0.072) did not exhibit a significant nonlinear inverse association with frailty.

## Discussion

Our study, while finding an overall non-significant association between the intake of various anthocyanidins and total anthocyanidins and the onset of frailty in adults, revealed a notable risk reduction in frailty with higher consumption of most anthocyanidins and total anthocyanidins. This protective effect was particularly evident in individuals aged 41–60 years and males. The research breaks new ground by exploring the link between diverse anthocyanidins and frailty across the general population. Moreover, our study investigated dose-response relationships in this context. These findings enhance our understanding of the potential benefits of anthocyanidins in mitigating frailty, especially among middle-aged individuals and males.

Flavonoids, a group of polyphenolic compounds widely present in plant foods like fruits, vegetables, tea, cocoa, and wine [29], exert various biological effects, including antioxidant, anti-inflammatory, anti-apoptotic, and neuroprotective activities [30]. Previous studies have investigated the association between flavonoid intake and frailty in various populations. For instance, Guo et al. reported U-shaped associations between the intake of total flavonoids—as well as several subclasses including flavan-3-ols, flavanones, flavones, flavonols, and isoflavones—and frailty among middle-aged and older adults (≥50 years) in the United States [10]. Similarly, Cai et al. found that higher consumption of flavonoids, particularly anthocyanidins, flavanones, flavones, and flavonols, was significantly associated with a lower prevalence of frailty in adults aged 20 years and older [11]. However, the relationship between anthocyanidins and their subtypes and frailty risk among U.S. adults aged 18 years and above remains unclear.

Among flavonoid subclasses, anthocyanidins have garnered attention for their potential role in preventing frailty. Studies have shown that anthocyanin-rich foods may affect the gut microbiome's composition, acting as mediators for positive health

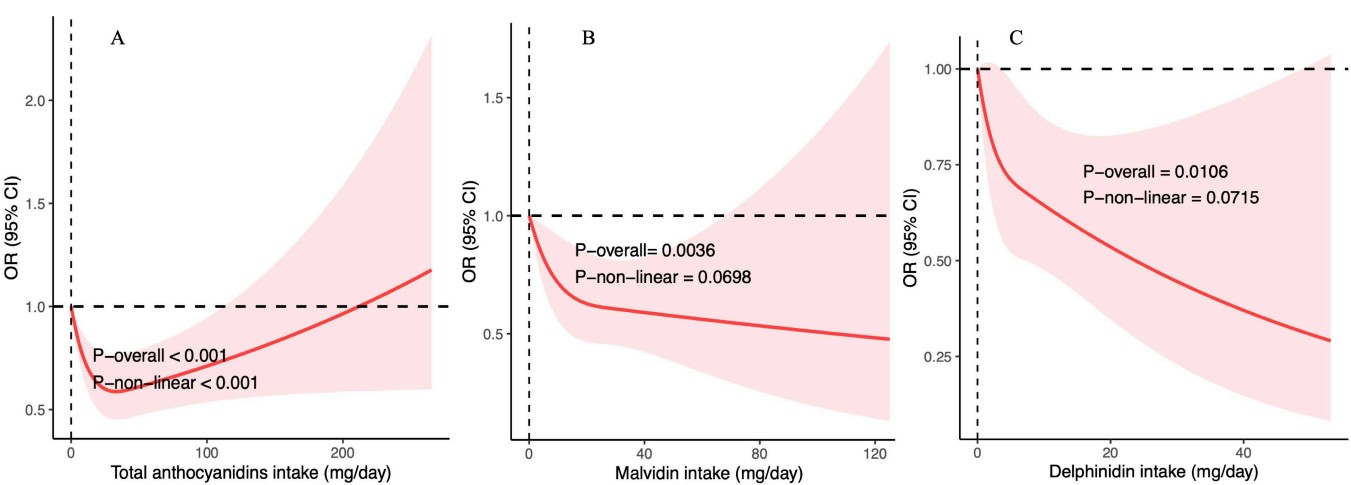

**Fig 2. Restricted cubic spline models illustrating the associations between dietary intake and frailty for: (a) total anthocyanidins, (b) malvidin, and (c) delphinidin.** The red lines represent the estimated log odds ratios (ORs), and the shaded areas indicate the corresponding 95% confidence intervals (CIs). The reference point was set at the lowest level of daily intake (0 mg/day) for each respective compound.

outcomes. These compounds undergo digestion, forming metabolites with widespread distribution in the body, contributing to beneficial biological effects. These compounds can be digested by various structures in the gut to form metabolites that are distributed throughout the body and have positive biological effects [31]. Despite the potential benefits of anthocyanidins, the underlying mechanism for their protective effects against frailty is not yet fully understood. However, several hypotheses have been proposed to explain their observed effects. One hypothesis is that anthocyanidins can act as antioxidants, scavenging reactive oxygen and nitrogen species, chelating transition metals, and modulating endogenous antioxidant enzymes and pathways [32,33]. Another hypothesis is that anthocyanidins can act as anti-inflammatory agents, inhibiting the activation of nuclear factor-kappa B (NF-κB) and the production of pro-inflammatory cytokines, such as tumor necrosis factor-alpha (TNF-α) and interleukin-6 (IL-6), which are elevated in frail individuals [34]. A third hypothesis is that anthocyanidins can act as neuroprotective agents, preventing neuronal death and dysfunction caused by excitotoxicity, which is mediated by excessive stimulation of glutamate receptors, such as N-methyl-D-aspartate (NMDA) receptors [35]. Anthocyanidins can cross the blood-brain barrier and reach the brain, where they may exert neuroprotective effects against oxidative stress, inflammation, and excitotoxicity, which are implicated in the pathogenesis of frailty and its related cognitive impairment [36]. Anthocyanidins may also modulate other cellular signaling pathways and gene expression involved in synaptic plasticity, neurogenesis, and cognition [37,38]. Our study demonstrated a reversal association between total anthocyanidin consumption at or above the median and frailty. However, only the intake of malvidin and delphinidin at or above median was significantly inversely associated with frailty when we examined individual anthocyanidins. Interestingly, this inverse association was significant for all anthocyanidins except pelargonidin when we excluded participants with co-morbid conditions, such as stroke, coronary heart disease, and depressive symptoms. This indicates that co-morbid conditions may obscure the protective effect of anthocyanidins on frailty.

Moreover, our subgroup analysis revealed a significant association between anthocyanidin consumption and a lower risk of frailty exclusively within the 41-60 age group. Although frailty is more evident in older adults, many frailty parameters are already altered in middle-aged individuals [39]. Frailty is a dynamic condition that deteriorates with age, implying that it is vital to screen and prevent frailty in middle-aged individuals, as they have a higher chance of recovering from frailty to robustness than older adults [40]. This may account for the significant inverse relationship between anthocyanidin consumption and frailty in middle-aged adults. Notably, physical well-being typically peaks between the ages of 18 and 40, potentially accounting for the lack of discernible protective effects of anthocyanidins in this age group. Contrarily, the intake of anthocyanidins at or above the median did not exhibit a risk reduction for frailty in adults aged 60 and above. One possible explanation is that older adults tend to have higher levels of oxidative stress and systemic inflammation [41]. These elevated levels of free radicals and inflammatory markers may exceed the antioxidant and anti-inflammatory capacity of dietary anthocyanidins, thus attenuating their protective effects in this population. Our subgroup analysis also revealed the significant between anthocyanidins consumption in male rather in female. The true reason is far from clear, and the difference in the metabolism, bioavailability, and physiological response to flavonoids may contribute to the stronger association observed in males. For instance, males may have higher baseline oxidative stress or inflammatory burden [42], making them more susceptible to the beneficial effects of anthocyanidins. Hormonal differences, such as the modulating role of estrogen in females, may also influence the efficacy of flavonoid-related pathways [43].

Determining the optimal level of total anthocyanidin consumption is paramount, as both insufficient and excessive intake can result in nutritional imbalances. Currently, there is no officially established recommended daily intake for anthocyanidins. A previous observational study reported a U-shaped association between total flavonoid intake and frailty, with the lowest risk observed at approximately 220 mg/day. In that study, the association between anthocyanidin intake and frailty was described as linear, making it difficult to identify a specific optimal dose [10]. However, it is important to note that the prior study focused solely on individuals aged 50 years and older, whereas our analysis included a broader adult population starting from 18 years of age. Based on our findings, the optimal intake of total anthocyanidins for reducing frailty risk is approximately 33.322 mg per day. Inadequate anthocyanidin consumption, indicative of a scarcity of fruits and vegetables, may lead to a shortfall in essential nutrients. Conversely, an abundance of anthocyanidins, signifying an

excessive intake of fruits and vegetables, could impede the consumption of adequate calories and other vital nutrients like protein. This imbalance may contribute to a diet of poor quality, elevating the risk of frailty. Achieving an appropriate and balanced level of anthocyanidin intake is crucial to maintaining a nutritionally sound diet and minimizing the risk of frailty.

Our study exhibits various strengths. Firstly, it investigated the relationship between the intake of distinct anthocyanidin types and frailty across different age groups and genders in the general adult population. This provides valuable insights for shaping public health recommendations regarding specific dietary preferences. Secondly, it employed dose–response and subgroup analyses, offering a more nuanced understanding of the associations and identifying potential subgroups that might derive benefits from consuming specific anthocyanidin. Thirdly, the study benefited from a large and nationally representative sample of American adults, enhancing the statistical power and reliability of the findings. We meticulously accounted for various confounding factors, including age, gender, and other dietary variables, fortifying the validity of our results. Additionally, the inclusion of diverse age groups enabled the exploration of potential age-specific effects, unveiling the notable protective impact of anthocyanidins in individuals aged 41-60 years. Notably, our study is the first to investigate the correlation between specific anthocyanidin types and the frailty index, a comprehensive measure combining laboratory and clinical assessments. The frailty index, being a continuous measure, proves advantageous for longitudinal studies as it allows for a more precise quantification of health status progression over time.

Despite these strengths, our study has several limitations. Firstly, the cross-sectional design of the NHANES data limits our ability to establish causality between anthocyanidin consumption and frailty risk. Longitudinal studies would be beneficial in determining the temporal relationship between anthocyanidin intake and frailty onset. Additionally, the reliance on self-reported dietary data may introduce recall bias and inaccuracies in estimating anthocyanidin intake. Future studies utilizing more objective measures, such as biomarkers or dietary records, would provide more accurate data on anthocyanidin consumption. Lastly, our study focused on the overall population, and we acknowledge that there may be variations in the effects of anthocyanidins on frailty risk among different subgroups, such as individuals with specific health conditions or genetic predispositions. Further research is needed to explore these potential subgroup differences.

## Conclusion

In conclusion, our study found a higher intake of most types of anthocyanidins and total anthocyanidins was associated with a reduced risk of developing frailty. This protective effect was particularly prominent in individuals aged 41–60 years and males. These findings contribute to our understanding of the potential benefits of anthocyanidins in mitigating frailty, especially among middle-aged individuals and males. However, further research is warranted to elucidate the underlying mechanisms and validate these findings in larger and more diverse populations.

## Supporting information

**S1 Table. Variables in the 49-item frailty index and their respective scorings.**
(XLSX)

**S2 Table. Relationship between different anthocyanidins consumption levels and frailty after excluding participants with stroke, coronary heart disease, and depressive symptom.**
(DOCX)

## Acknowledgments

The present study utilizes data sourced from the National Health and Nutrition Examination Survey (NHANES). We express our gratitude to the National Center for Health Statistics (NCHS) and the United States Centers for Disease Control and Prevention (CDC) for their generous funding assistance in facilitating the gathering and analysis of data. We express our gratitude to both the study participants and the staff involved in this research endeavor.

## Author contributions

**Conceptualization:** Xiaofeng Zhang, Yanfei Wu.

**Data curation:** Xiaofeng Zhang.

**Formal analysis:** Xiaofeng Zhang, Zhenhua Jin, Yanfei Wu.

**Funding acquisition:** Xiaofeng Zhang, Yanfei Wu.

**Investigation:** Xiaofeng Zhang, Pengpeng Huang, Yanfei Wu.

**Methodology:** Xiaofeng Zhang, Yanfei Wu.

**Project administration:** Yanfei Wu.

**Resources:** Xiaofeng Zhang.

**Software:** Xiaofeng Zhang, Zhenhua Jin, Yanfei Wu.

**Supervision:** Yanfei Wu.

**Validation:** Xiaofeng Zhang, Yanfei Wu.

**Visualization:** Yanfei Wu.

**Writing – original draft:** Xiaofeng Zhang.

**Writing – review & editing:** Zhenhua Jin, Yanfei Wu.

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
