## [Decision Letter · Decision Letter 0]

Dear Dr. Wu,

Thank you for submitting your manuscript to PLOS ONE. After careful consideration, we feel that it has merit but does not fully meet PLOS ONE’s publication criteria as it currently stands. Therefore, we invite you to submit a revised version of the manuscript that addresses the points raised during the review process.

**We have now received the reviewers' comments and suggestions, and we should address them thoroughly and resubmit the manuscript to the journal for further consideration.**

We look forward to receiving your revised manuscript.

Kind regards,

Roshan Thotagamuge, PhD

Academic Editor

PLOS ONE

**Journal Requirements:**

1. When submitting your revision, we need you to address these additional requirements. Please ensure that your manuscript meets PLOS ONE's style requirements, including those for file naming. The PLOS ONE style templates can be found at https://journals.plos.org/plosone/s/file?id=wjVg/PLOSOne_formatting_sample_main_body.pdf and https://journals.plos.org/plosone/s/file?id=ba62/PLOSOne_formatting_sample_title_authors_affiliations.pdf 2. Thank you for stating in your Funding Statement: This work was supported by the Medical Science and Technology Project of Zhejiang Province(No. 2022KY576,No.2023KY453) Please provide an amended statement that declares *all* the funding or sources of support (whether external or internal to your organization) received during this study, as detailed online in our guide for authors at http://journals.plos.org/plosone/s/submit-now.  Please also include the statement “There was no additional external funding received for this study.” in your updated Funding Statement. Please include your amended Funding Statement within your cover letter. We will change the online submission form on your behalf. 3. Thank you for stating the following in the Acknowledgments Section of your manuscript: The present study utilizes data sourced from the National Health and Nutrition Examination Survey (NHANES). We express our gratitude to the National Center for Health Statistics (NCHS) and the United States Centers for Disease Control and Prevention (CDC) for their generous funding assistance in facilitating the gathering and analysis of data. We express our gratitude to both the study participants and the staff involved in this research endeavor. We note that you have provided funding information that is not currently declared in your Funding Statement. However, funding information should not appear in the Acknowledgments section or other areas of your manuscript. We will only publish funding information present in the Funding Statement section of the online submission form. Please remove any funding-related text from the manuscript and let us know how you would like to update your Funding Statement. Currently, your Funding Statement reads as follows: This work was supported by the Medical Science and Technology Project of Zhejiang Province(No. 2022KY576,No.2023KY453)  Please include your amended statements within your cover letter; we will change the online submission form on your behalf. 4. Thank you for stating the following in your Competing Interests section:  All authors disclosed no relevant relationships. Please complete your Competing Interests on the online submission form to state any Competing Interests. If you have no competing interests, please state "The authors have declared that no competing interests exist.", as detailed online in our guide for authors at http://journals.plos.org/plosone/s/submit-now This information should be included in your cover letter; we will change the online submission form on your behalf. 5. Please note that your Data Availability Statement is currently missing the DOI/accession number of each dataset OR a direct link to access each database. If your manuscript is accepted for publication, you will be asked to provide these details on a very short timeline. We therefore suggest that you provide this information now, though we will not hold up the peer review process if you are unable.

Reviewers' comments:

Reviewer's Responses to Questions

**Comments to the Author**

1. Is the manuscript technically sound, and do the data support the conclusions?

Reviewer #1: Yes

Reviewer #2: Yes

2. Has the statistical analysis been performed appropriately and rigorously?

Reviewer #1: I Don't Know

Reviewer #2: Yes

3. Have the authors made all data underlying the findings in their manuscript fully available?

Reviewer #1: Yes

Reviewer #2: No

4. Is the manuscript presented in an intelligible fashion and written in standard English?

Reviewer #1: Yes

Reviewer #2: Yes

**Reviewer #1: ** Dear Author,

Remarks to the author:

This study offers important insights into the link between dietary anthocyanidins and reduced frailty risk, especially in middle-aged adults and males. The use of NHANES data strengthens the findings, and the identified U-shaped dose-response relationship is particularly noteworthy. While the results are promising, the cross-sectional design limits causal conclusions, highlighting the need for further longitudinal research.

Specific comments:

1) Please clarify whether the study population is representative of the entire U.S. population or specific to a particular region within the country.

2) The study concludes a potential protective role of dietary anthocyanidins in reducing frailty, particularly among middle-aged individuals and males. Could the authors elaborate on why this effect is more pronounced in males compared to females, and why it appears significant in middle age but not in older adults?

3) As the authors mention, flavonoids are found in plant-based products such as tea, coffee, and wine. However, excessive consumption of these foods may not be beneficial. Is there a recommended daily intake of flavonoids for different age groups and genders?

Thanks.

**Reviewer #2:**  This study addresses an important topic and is generally interesting. However, there are some gaps that need to be addressed for clarity and rigor. The comments were provided to improve the quality of the manuscript.

**Do you want your identity to be public for this peer review?** For information about this choice, including consent withdrawal, please see our Privacy Policy

Reviewer #1: No

Reviewer #2: No

---

## [Author Response · Author response to Decision Letter 1]

15 Jun 2025

Thank you very much for your constructive comments and suggestions, which have significantly improved the quality of our work.

We have carefully corrected all the mistakes pointed by reviewers and editor. Please see these modifications in the attached documents (Response to Reviewers.pdf) in detail.

---

## [Decision Letter · Decision Letter 1]

The inverse relationship between dietary anthocyanidins consumption and Frailty: Findings from the National Health and Nutrition Examination Survey

PONE-D-25-11030R1

Dear Dr. Wu,

We’re pleased to inform you that your manuscript has been judged scientifically suitable for publication and will be formally accepted for publication once it meets all outstanding technical requirements.

Kind regards,

Roshan Thotagamuge, PhD

Academic Editor

PLOS ONE

Additional Editor Comments (optional):

Following the revision, the manuscript has improved and is suitable for publication in the journal, provided that the minor amendments suggested by Reviewer 2 are addressed.

Reviewers' comments:

Reviewer's Responses to Questions

**Comments to the Author**

Reviewer #1: All comments have been addressed

Reviewer #2: All comments have been addressed

2. Is the manuscript technically sound, and do the data support the conclusions?

Reviewer #1: Yes

Reviewer #2: Yes

3. Has the statistical analysis been performed appropriately and rigorously?

Reviewer #1: N/A

Reviewer #2: Yes

4. Have the authors made all data underlying the findings in their manuscript fully available?

Reviewer #1: Yes

Reviewer #2: No

5. Is the manuscript presented in an intelligible fashion and written in standard English?

Reviewer #1: Yes

Reviewer #2: Yes

Reviewer #1: (No Response)

Reviewer #2: I appreciate authors’ efforts in revising the manuscript. Most of the comments have been addressed satisfactorily, though a few minor points still require attention.

Materials and Methods:

- Regarding “To derive the frailty index value, the participant's acquired deficits' number is divided by the total potential deficits.”: According to previously suggestion, authors have cited the primary sources for Frailty Index Calculation. Kindly check whether those are cited in the correct place.

Results:

- Regarding Table 1:

i. N and n should be defined at the bottom of table 1.

ii. Authors have stated that the presented percentages are based on survey-weighted analyses using NHANES sampling weights. To enhance transparency and interpretability for readers, it may be helpful to also report the weighted sample size. Including both the unweighted and weighted numbers, either in the table or as a footnote, would provide clarity regarding the findings.

iii. The authors have included effect size metrics as suggested. It is appreciated. However, they have not provided any interpretation of these metrics, in terms of their relevance within the results section.

- Regarding Table 2: The letter 'e' in the table footnote should be bolded for consistency with the formatting of other footnote indicators.

- Although the authors have mentioned the supplementary tables, these have not yet been attached to the manuscript.

References:

- A few references are still not formatted uniformly. Please review and revise the reference list to ensure consistency in formatting, adhering to the required citation style.

**Do you want your identity to be public for this peer review?** For information about this choice, including consent withdrawal, please see our Privacy Policy

Reviewer #1: No

Reviewer #2: No
